# Mucosal-associated invariant T (MAIT) cells mediate protective host responses in sepsis

Shubhanshi Trivedi[1], Daniel Labuz[1], Cole P Anderson[1], Claudia V Araujo[2], Antoinette Blair[2], Elizabeth A Middleton[2,3], Owen Jensen[1], Alexander Tran[4], Matthew A Mulvey[4], Robert A Campbell[2,5], J Scott Hale[4], Matthew T Rondina[2,5,6†], Daniel T Leung[1,4†]*

[1]Division of Infectious Diseases, University of Utah, Salt Lake City, United States; [2]Molecular Medicine Program, University of Utah, Salt Lake City, United States; [3]Division of Pulmonary and Critical Care, University of Utah, Salt Lake City, United States; [4]Division of Microbiology and Immunology, Department of Pathology, University of Utah, Salt Lake City, United States; [5]Division of General Internal Medicine, Department of Internal Medicine, University of Utah, Salt Lake City, United States; [6]George E. Wahlen VAMC Department of Internal Medicine and GRECC, University of Utah, Salt Lake City, United States

**Abstract** Sepsis is a systemic inflammatory response to infection and a leading cause of death. Mucosal-associated invariant T (MAIT) cells are innate-like T cells enriched in mucosal tissues that recognize bacterial ligands. We investigated MAIT cells during clinical and experimental sepsis, and their contribution to host responses. In experimental sepsis, MAIT-deficient mice had significantly increased mortality and bacterial load, and reduced tissue-specific cytokine responses. MAIT cells of WT mice expressed lower levels of IFN-γ and IL-17a during sepsis compared to sham surgery, changes not seen in non-MAIT T cells. MAIT cells of patients at sepsis presentation were significantly reduced in frequency compared to healthy donors, and were more activated, with decreased IFN-γ production, compared to both healthy donors and paired 90-day samples. Our data suggest that MAIT cells are highly activated and become dysfunctional during clinical sepsis, and contribute to tissue-specific cytokine responses that are protective against mortality during experimental sepsis.

*For correspondence:
daniel.leung@utah.edu

†These authors contributed equally to this work

Competing interests: The authors declare that no competing interests exist.

## Introduction

Sepsis is a life-threatening syndrome caused by dysregulated host responses to infection resulting in organ dysfunction (*Singer et al., 2016*). It is a leading cause of death worldwide, accounting for 5.3 million deaths annually (*Fleischmann et al., 2016*). Sepsis is characterized by a systemic acute pathologic hyper-inflammatory response and an opposing anti-inflammatory response that can occur concurrently (*Gentile et al., 2012*). Hallmarks of sepsis include a decreased ability to eliminate primary pathogens and an increased susceptibility to secondary nosocomial infections (*Otto et al., 2011*). In addition, it is associated with a protracted immunosuppressed state that contributes to morbidity and mortality (*Hotchkiss et al., 2013*). Unfortunately, clinical trials over the past 3 decades of interventions targeting the hyper-inflammatory state of sepsis have largely failed to consistently improve clinical outcomes (*Cohen et al., 2012*; *PROWESS-SHOCK Study Group et al., 2012*; *Williams, 2012*), and effective treatments are needed.

Many components of host responses are altered during sepsis, including activation of innate effector cells such as platelets, neutrophils, epithelial cells, and endothelial cells. In some settings, cellular activation is associated with dysfunctional responses and adverse clinical outcomes. Studies on immunosuppression in sepsis have largely focused on exhaustion, apoptosis, and reprogramming of a broader range of effector cells, including T cells, B cells, neutrophils, and other antigen-presenting cells (*Hotchkiss et al., 2016*). Unconventional T cells, such as invariant NKT (iNKT) cells, γδ T cells, and mucosal-associated invariant T (MAIT) cells, bridge the innate and adaptive arms of the immune response, and can contribute to both early-phase inflammation and late-phase immunosuppression.

MAIT cells are innate-like T cells restricted by MHC-related molecule 1 (MR1) and MAIT ligands belong to a class of transitory intermediates of the riboflavin synthesis pathway (*Porcelli et al., 1993*). They express an invariant T cell receptor (TCR) TRAV1-2 (or Vα7.2) in humans and TRAV-1 (or Vα19) in mice and a variable, but restricted, number of TRAJ and TCR β chains. MAIT cells are abundant in mucosal tissues such as the liver, lung, and mesenteric lymph nodes and gut lamina propria. In humans, they constitute 1–10% of peripheral blood T lymphocytes, up to 10% of intestinal T cells, and up to 40% of T cells in the liver (*Lee et al., 2014*; *Martin et al., 2009*). Recent studies have highlighted the protective role of MAIT cells in host antibacterial responses in vivo (*Georgel et al., 2011*; *Chua et al., 2012*; *Le Bourhis et al., 2010*). It has also been shown that in patients with sepsis, MAIT cell frequencies are decreased in circulation compared to healthy control donors and uninfected critically-ill patients (*Grimaldi et al., 2014*). Septic patients with persistent MAIT cell depletion also have a higher incidence of secondary, ICU-acquired infections (*Grimaldi et al., 2014*). However, detailed phenotypic and functional MAIT cell changes during sepsis, as well as the mechanisms by which MAIT cells contribute to host immune responses in sepsis, are not known. In this work, we used complementary, longitudinal studies in sepsis patients and in a relevant murine sepsis model to study the role of MAIT cells in sepsis pathology. We examined the immune responses in C57BL/6 wild type (WT) or MR1 knock-out (*Mr1⁻/⁻*; i.e. MAIT depleted) mice using the cecal ligation and puncture (CLP) model of polymicrobial sepsis. Additionally, we evaluated the phenotype and function of human MAIT cells during acute sepsis and at 3 months after sepsis.

## Results

### MAIT cells have decreased expression of effectors during experimental sepsis

To assess MAIT cell effector function during sepsis, we examined MAIT cells in the lungs of WT mice undergoing CLP, a polymicrobial model of sepsis, or sham surgery. We chose lungs given the frequent association of lung injury with clinical sepsis (*Fein et al., 1983*; *Martin et al., 2003*), and because it is among the most abundant site of MAIT cells in mice (*Rahimpour et al., 2015*). While we noted similar frequencies of MR1-5OP-RU-tetramer⁺ (from here on, referred to as MR1-tetramer⁺) MAIT cells in CLP mice as in sham mice (*Figure 1A and B*), we found in sorted MAIT cells that gene expression of MAIT effectors *Ifng* (p=0.04) and *Il17a* (p=0.02) were significantly lower (and *GzmB* trended lower, p=0.14) in septic mice compared to sham mice (*Figure 1C*). Conversely, in non-MAIT TCRβ⁺CD3⁺ T cell populations, *Ifng* (p=0.06), *Il17a* (p=0.16), and *Gzmb* (p=0.49) mRNA expression levels were similar between septic mice and sham mice (*Figure 1D*).

To further confirm that MAIT cell effector function is impaired during experimental sepsis, we evaluated IFNγ, TNFα, IL-17a, GM-CSF, and IL-10 protein expression in MAIT cells of lung tissue using flow cytometry. We found that similar to *Ifng* mRNA expression, percentage frequencies of MAIT cells expressing IFNγ were significantly lower (p=0.04) in septic mice compared to sham mice (*Figure 2A*). The mean fluorescent intensity (MFI) of IFNγ staining in TCRβ⁺ MR1-5OP-RU-tetramer⁺ MAIT cells was also significantly reduced (p=0.02) in septic mice compared to sham mice (*Figure 2B*). Conversely, in non-MAIT TCRβ⁺CD3⁺ T cell populations, IFNγ protein expression was significantly higher (p=0.0005) in septic mice compared to sham mice (*Figure 2C*). No significant differences were observed in IFNγ expression in *i*NKT and TCRγδ cells (*Figure 2D and E*). No significant differences were observed in frequencies of TNFα, IL-17a, GM-CSF, and IL-10 between the groups.

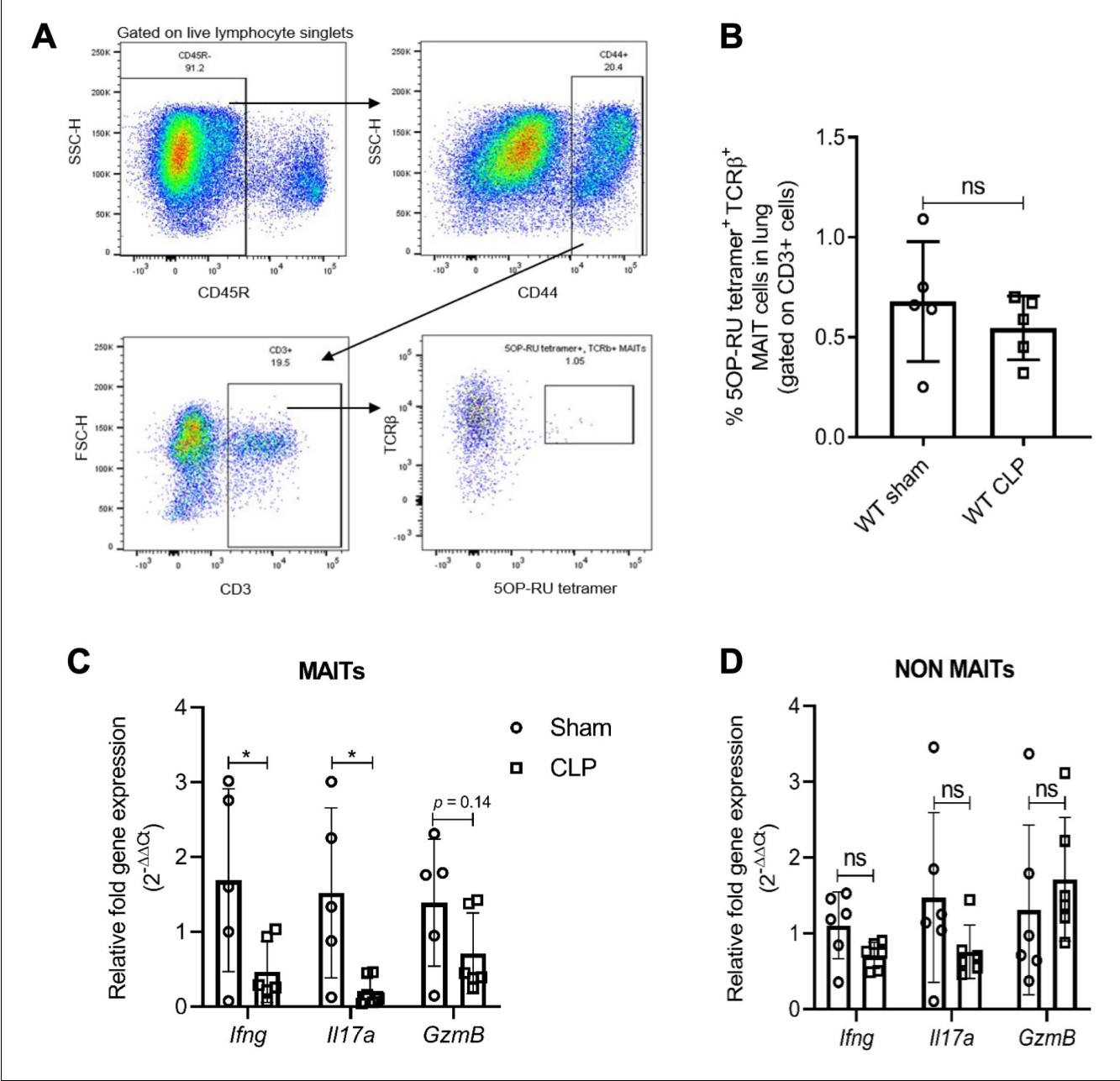

**Figure 1.** Sepsis induces MAIT-specific changes in inflammatory cytokine expression. (A) Gating strategy for isolation of MAIT cells. In the first plot, lymphocytes with CD45R⁻ B cells were excluded by electronic gating and CD44⁺ CD3⁺ T cells were gated to define MAIT cells as TCRβ⁺ MR1-5OP-RU-tetramer⁺ cells. (B) Percentage of TCRβ⁺ MR1-tetramer⁺ MAIT cells, of CD3⁺ T cells in WT mice after 18 hr of CLP or sham operation. (C) Gene expression in MAIT cells isolated using flow cytometric sorting of homogenized lung tissue after CLP or sham operation (n = 5 per group), as determined by qRT-PCR for *Ifng*, *Il17a*, and *GzmB* genes, (D) *Ifng*, *Il17a*, and *GzmB* mRNA expression in non-MAIT (MR1-tetramer⁻ TCRβ⁺) CD3⁺ T cell populations. Data are expressed as mean ± SD, and an unpaired t-test was used for comparisons (data passed the Shapiro-Wilk normality test).

## MAIT deficiency increases bacterial burden and mortality during experimental sepsis

We examined whether the absence of MAIT cells altered survival outcomes in vivo during experimental sepsis. In the CLP model of polymicrobial sepsis (*Yost et al., 2016*; *Middleton et al., 2019*), we saw that MR1-deficient mice (*Mr1⁻/⁻*) which lack MAIT cells, had significantly increased sepsis-related mortality compared to WT mice (*Figure 3A*). The majority (11/15,~73%) of *Mr1⁻/⁻* mice died 24 to 48 hr following induction of sepsis, while the majority (13/15,~87%) of WT mice survived up to

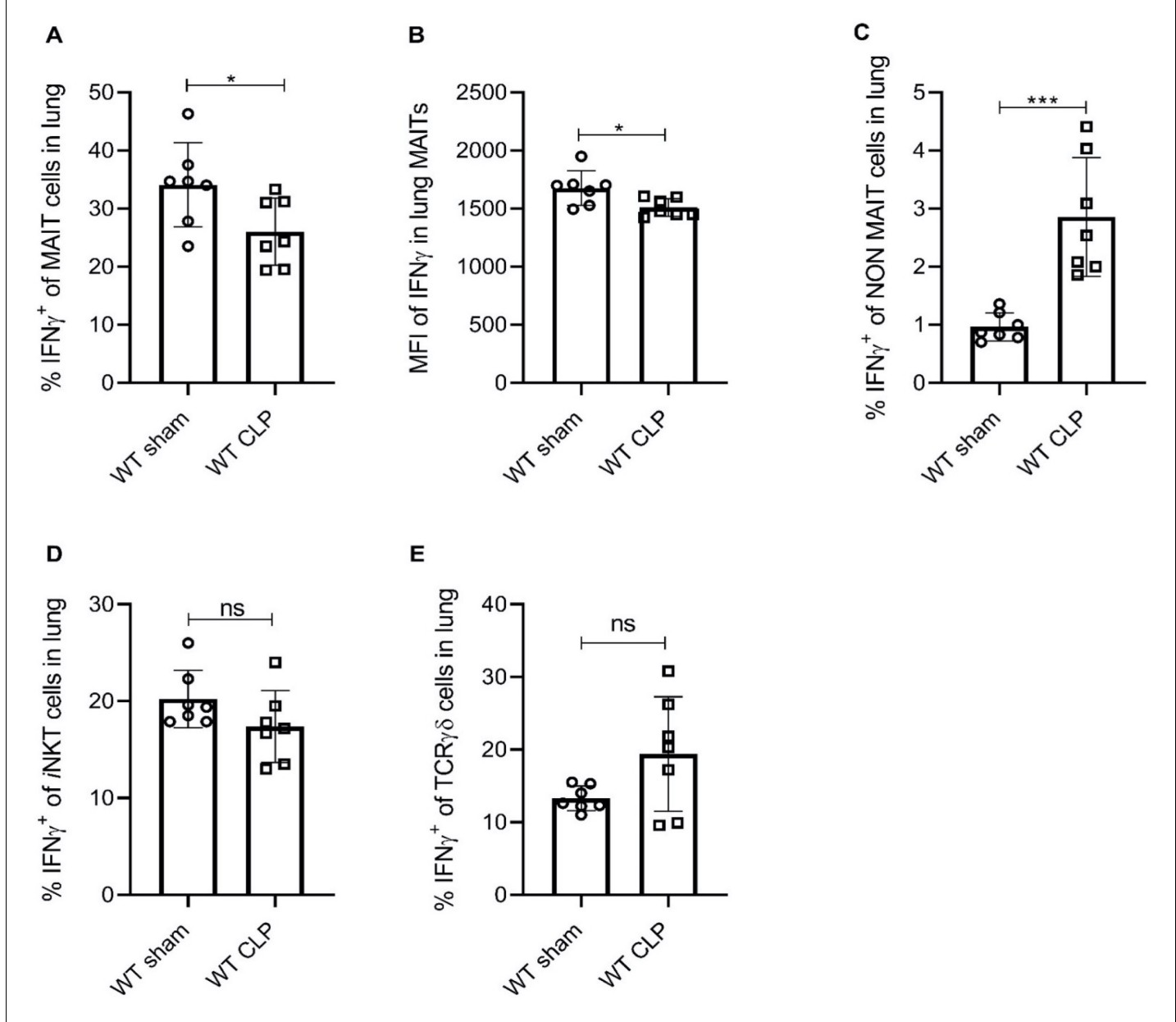

**Figure 2.** Sepsis induces lower IFNγ production in MAIT cells. (A) Percentage frequencies of IFNγ expression in MAIT cells of lung tissue, 18 hr after CLP or sham operation (n = 7 per group), as determined by intracellular cytokine staining and flow cytometry. (B) MFI of IFNγ staining in MAIT cells. (C) Percentage frequencies of IFNγ expression in non-MAIT cells of lung tissue. (D and E) Percentage frequencies of IFNγ expression in iNKT and TCRγδ cells. Data are expressed as mean ± SD, and an unpaired t-test was used for comparisons (data passed the Shapiro-Wilk normality test).

100 hr after sepsis (*Figure 3A*). On examining the bacterial counts 18 hr following CLP, we found that $Mr1^{-/-}$ mice had a significantly higher burden of bacteria in blood compared to WT mice (*Figure 3B*). The microbiota associated with $Mr1^{-/-}$ mice may differ from WT mice (*Smith et al., 2019*); therefore, we examined whether such differences affect susceptibility to sepsis. In experiments in which bedding was transferred between $Mr1^{-/-}$ and WT cages (*Miyoshi et al., 2018*) prior to CLP, we found that $Mr1^{-/-}$ mice had significantly increased mortality compared to WT mice (*Figure 3C*). To extend these findings to a setting of sepsis due to a single pathogen, we induced sepsis via intraperitoneal injection of a clinical strain of extraintestinal pathogenic *Escherichia coli* (ExPEC). As we observed in the CLP, polymicrobial model of sepsis, $Mr1^{-/-}$ mice had significantly higher mortality from ExPEC sepsis, compared to WT mice (*Figure 3—figure supplement 1*).

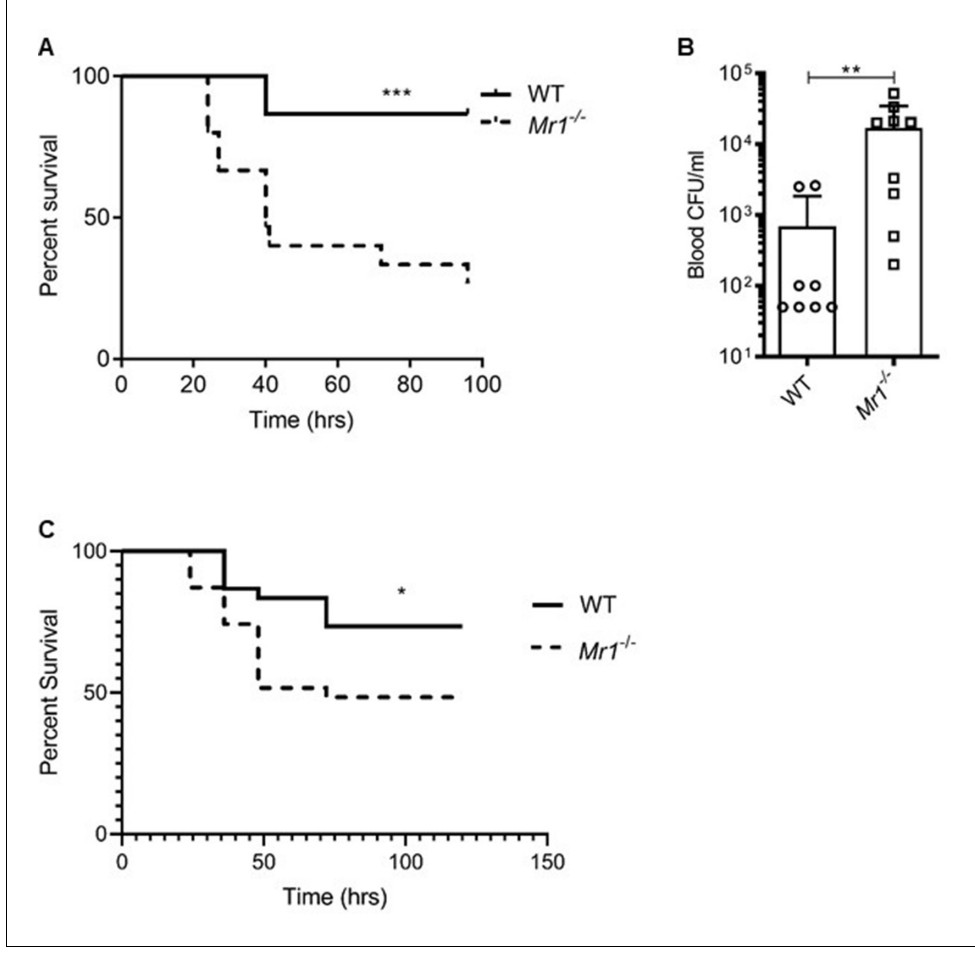

**Figure 3.** *Mr1*<sup>-/-</sup> mice had higher sepsis-induced mortality and bacterial burden compared to wild type mice. (**A**) Age-matched *Mr1*<sup>-/-</sup> and WT mice (n = 15 per group) underwent CLP to induce polymicrobial sepsis. Survival was recorded over a period of 4–5 days (WT versus *Mr1*<sup>-/-</sup> mice, ***p=0.0007). Data represent two independent experiments. Statistical analysis was performed using the Log-rank (Mantel-Cox) test. (**B**) Blood was collected from WT and *Mr1*<sup>-/-</sup> mice (n = 8–10 per group) 18 hr post-CLP, and serial dilutions were plated on Luria Broth (LB) agar plates. Colony-forming units (CFU) were determined 24 hr after plating and were expressed as CFU/mL. Statistical analysis was performed using the Mann-Whitney test. (**C**) The bedding transfer procedure (detailed in Materials and methods section) was used to exchange gut microbiome between age-matched *Mr1*<sup>-/-</sup> and WT mice (total n = 31 for *Mr1*<sup>-/-</sup> mice and total n = 30 for WT mice) before inducing sepsis by CLP. Survival was recorded over a period of 4–5 days (WT versus *Mr1*<sup>-/-</sup> mice, *p=0.03). Data represent three independent experiments. Statistical analysis was performed using the Log-rank (Mantel-Cox) test.

The online version of this article includes the following figure supplement(s) for figure 3:

**Figure supplement 1.** *Mr1*<sup>-/-</sup> mice had significantly higher mortality from ExPEC sepsis, compared to WT mice.

## MAIT deficiency is associated with reduced lung cytokine responses in experimental sepsis

To determine the contribution of MAIT cells to sepsis-induced tissue inflammatory responses, we evaluated MAIT-associated cytokines (*Xiao and Cai, 2017*) in lung homogenates after CLP. Compared to WT mice, *Mr1*<sup>-/-</sup> mice had significantly reduced levels of lung IFN-γ, TNFα, IL-17A, IL-10, and GM-CSF: cytokines produced by MAIT cells (*Xiao and Cai, 2017*; *Figure 4*). Interestingly, serum, liver, and gut cytokines were less affected than lung cytokines in *Mr1*<sup>-/-</sup> mice following sepsis (*Figure 4—figure supplements 1–3*).

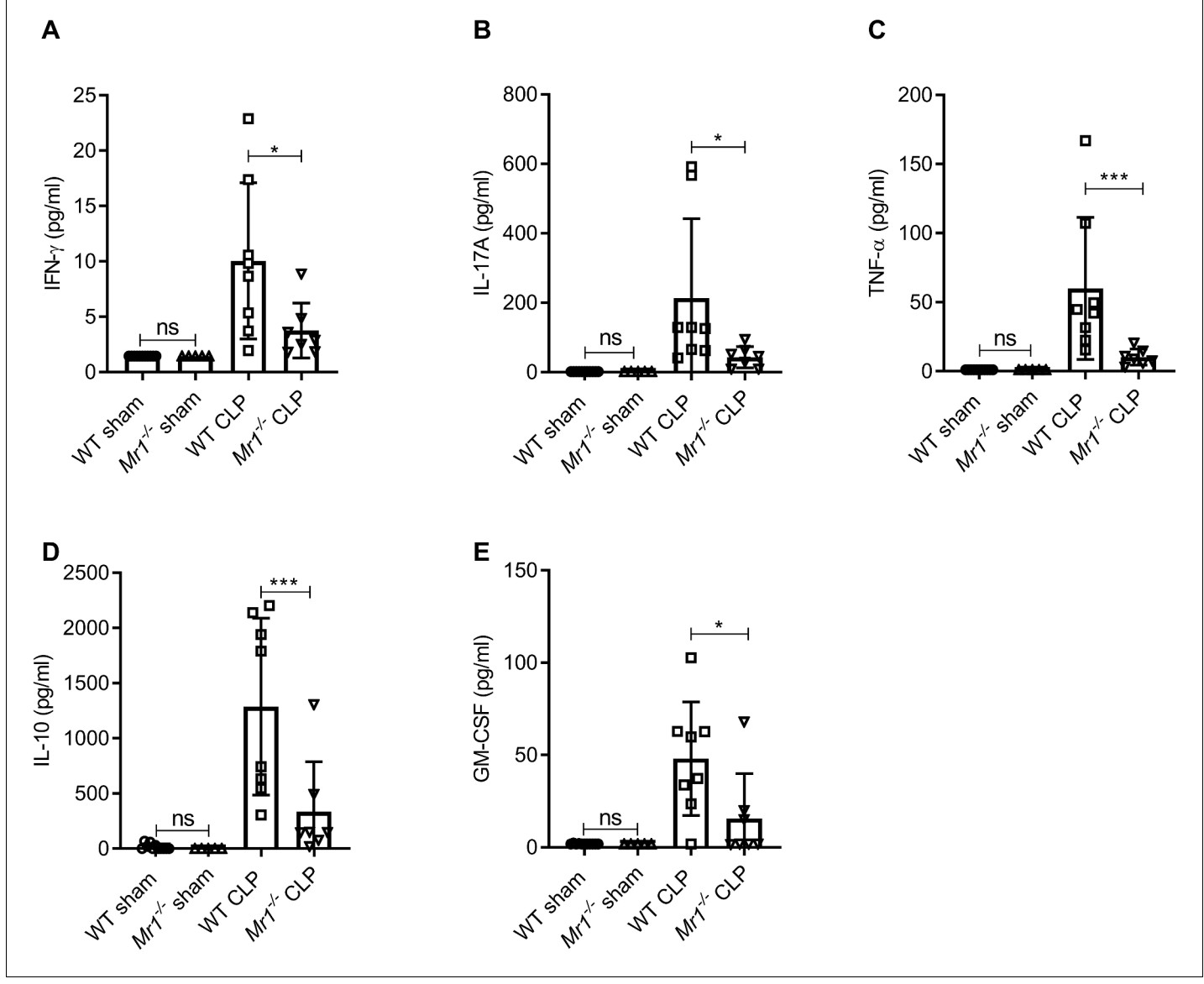

**Figure 4.** $Mr1^{-/-}$ mice had significantly lower levels of cytokines in the lung tissue at 18 hr post-CLP. Levels of (**A**) IFN-γ, (**B**) IL-17A, (**C**) TNF-α, (**D**) IL-10, and (**E**) GM-CSF were assessed in the lungs isolated from WT and $Mr1^{-/-}$ mice at 18 hr post-CLP or sham operation, using a bead-based immunoassay kit. Bars represent the mean cytokine levels ± SD. The graphs represent two independent experiments (WT n = 8, $Mr1^{-/-}$ n = 7). The statistical significance was determined by the nonparametric Mann–Whitney test.

The online version of this article includes the following figure supplement(s) for figure 4:

**Figure supplement 1.** $Mr1^{-/-}$ mice had reduced IFNγ and TNFα but similar levels of other cytokines in serum at 18 hr post-CLP.

**Figure supplement 2.** $Mr1^{-/-}$ mice had reduced IL-10 but similar levels of other cytokines in the liver at 18 hr post-CLP.

**Figure supplement 3.** $Mr1^{-/-}$ mice had similar levels of other cytokines in the gut at 18 hr post-CLP.

## MAIT deficiency is associated with reduced tissue-specific interstitial macrophages and monocytic dendritic cell frequencies

MAIT cells can contribute to cytokine release from macrophages (*Hegde et al., 2018*) and have shown to promote inflammatory monocyte differentiation into dendritic cells during pulmonary infection (*Meierovics and Cowley, 2016*). Thus, we next sought to determine the cause of reduced tissue cytokine production of $Mr1^{-/-}$ mice during sepsis. We approached this by determining how an absence of MAIT cells influences the frequencies of lung innate immune cells, including alveolar

macrophages (AMs), interstitial macrophages (IMs), monocytic dendritic cells (moDCs), and plasmocytoid DCs (pDCs). As compared to WT mice, $Mr1^{-/-}$ mice had a significantly lower proportion of IMs (Siglec-F[-], CD24[-], MHCII[+], CD11c[+], CD11b[+], CD64[+] cells; *Figure 5A*) and moDCs (Siglec-F[-], MHCII[+],-CD11c[+], CD11b[+], Ly-6C[+] cells; *Figure 5B*) following CLP. The loss of MAIT cells did not influence AMs, pDCs, or neutrophils, however, following CLP (*Figure 5C* -E). To investigate the possible mechanism by which MAIT cells are protective, we performed ex vivo expansion of lung MAIT cells from WT mice using 5-OP-RU/MR1 artificial antigen presenting cells (*Liu et al., 2020*) and co-cultured them with Ly6C[+] CD11b[+] monocytes, also isolated from WT mice, for 18 hr. We found significantly higher amounts of IFNγ (*Figure 5—figure supplement 1A*) and GM-CSF (*Figure 5—figure supplement 1B*) in co-cultures stimulated with 5-amino-6-D-ribitylaminouracil (5-A-RU) and methylglyoxal (MeG) (*Li et al., 2018*) compared to unstimulated and monocyte only controls indicating that MAIT cells in the lung can interact with monocytes to produce effector cytokines.

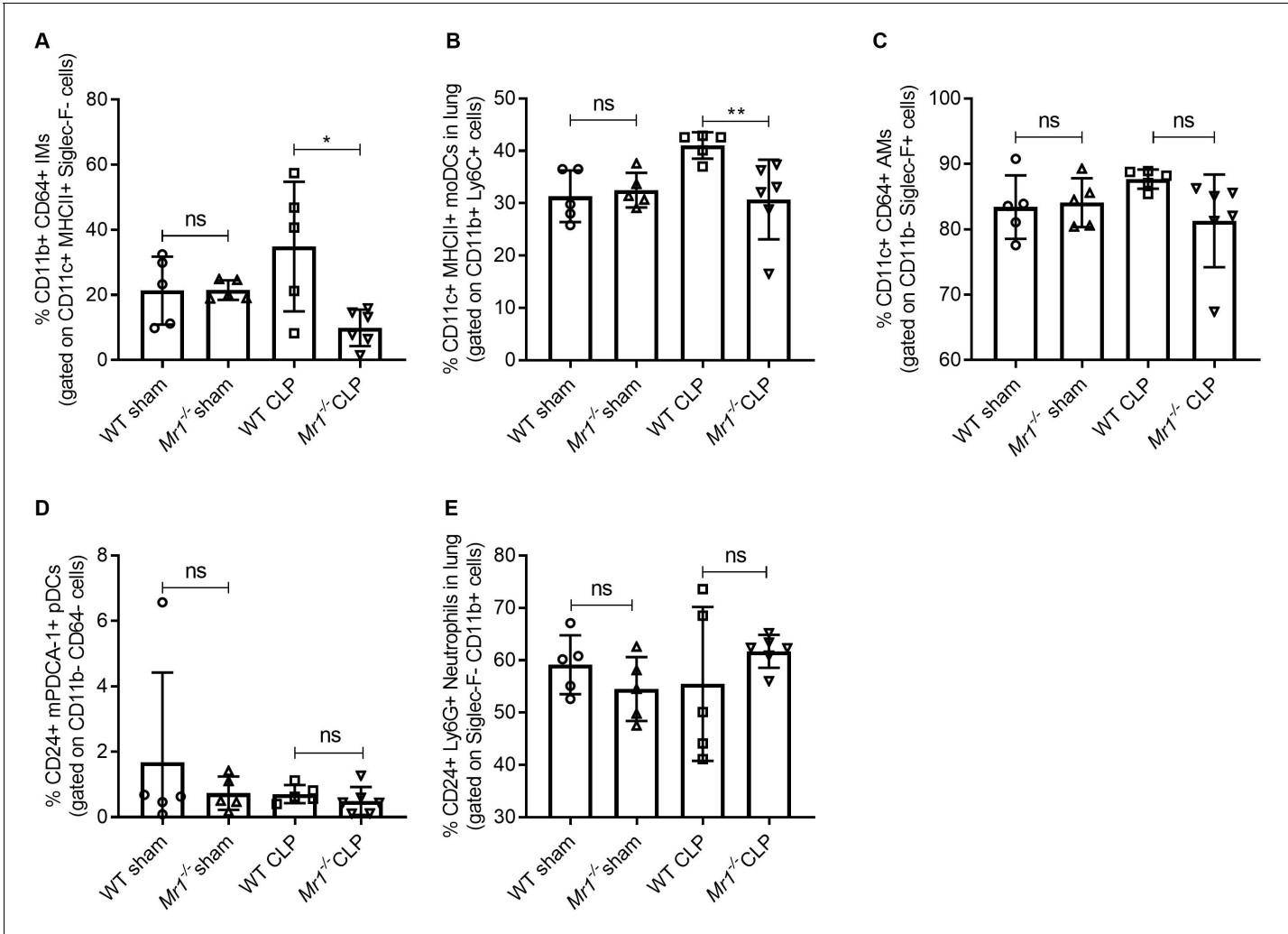

**Figure 5.** Lower frequencies of interstitial macrophages and monocytic dendritic cells in the lungs of $Mr1^{-/-}$ mice at 18 hr post-CLP. Lung cells were obtained from WT (n = 5) and $Mr1^{-/-}$ mice (n = 5–6) at 18 hr post-CLP or sham operation, stained for surface markers, and analyzed using flow cytometry. Percentage frequencies of (A) interstitial macrophages, (B) monocytic dendritic cells, (C) alveolar macrophages, (D) plasmacytoid dendritic cells, and (E) neutrophils were compared between the groups by the nonparametric Mann–Whitney test. Data were expressed as mean ± SD of two independent experiments.

The online version of this article includes the following figure supplement(s) for figure 5:

**Figure supplement 1.** Higher amounts of IFN-γ and GM-CSF production in ex vivo stimulated MAIT-monocytes coculture.

**Figure supplement 2.** Frequencies of other unconventional T cells such as invariant natural killer T cells (iNKT) and γδ T cells post-CLP.

Next, we examined whether the loss of MAIT cells altered the frequencies of other unconventional T cells such as invariant natural killer T cells (*i*NKT) (*Hu et al., 2009*; *Heffernan et al., 2013*) and γδ T cells (*Andreu-Ballester et al., 2013*; *Hirsh et al., 2004*) in sepsis. We observed that the frequencies of *i*NKT and γδ T cells in the lungs were reduced during sepsis, consistent with prior studies showing that frequencies of *i*NKT (*Hu et al., 2009*; *Heffernan et al., 2013*) and γδ T cells (*Andreu-Ballester et al., 2013*; *Hirsh et al., 2004*) are modulated in response to polymicrobial sepsis. However, there were no differences in the frequencies of *i*NKT or γδ T cells between $Mr1^{-/-}$ and WT mice (*Figure 5—figure supplement 2*).

## Consistent with murine sepsis studies, MAIT cell frequency, activation, and effectors are altered in septic patients

We examined MAIT cells isolated from the blood of 33 septic patients (16 female and 17 male, mean age 59 + / - 2.7 years, mean SOFA score 4.6) within 48 (±24) hr of ICU admission for sepsis (day 1, D1). In 12 of these same patients (6 female and 6 male, mean age 60.5 + / - 4 years), we repeated MAIT phenotyping again approximately 90 days after their ICU admission for sepsis. We compared them to 21 age- and sex-matched healthy donors (HD) (10 female and 11 male, mean age 51 + / - 3.8 years). The identified causes of sepsis included pneumonia, urinary tract, intra-abdominal, and soft-tissue infections. Upon ICU admission, septic patients showed lower frequencies of MR1-tetramer$^+$ MAIT cells within the T cell (CD3$^+$) compartment compared to healthy donors (HD) (*Figure 6*, A and B). At 90 days after sepsis, MAIT cell frequencies were not statistically different from Day 1 septic patients and HD (*Figure 6B and C*).

Next, we evaluated markers of MAIT cell activation, exhaustion, and intracellular cytokine expression in a subset of surviving septic patients and for which paired samples at both Day 1 and Day 90 were available (n = 12). Upon ICU admission (Day 1), the expression of MAIT cell activation markers, including CD69, CD38, and CD137, was significantly higher in septic patients compared to HD (*Figure 6D*). However, when assessed again in these same septic patients on Day 90, the expression of MAIT cell activation markers had returned to levels comparable to HD (*Figure 6D*). Upon ICU admission, we also found a higher expression of TIM-3 (p=0.03), and LAG-3 (trend with p=0.14), inhibitory receptors associated with T cell exhaustion, in MAIT cells of septic patients compared to HD (*Figure 6E*). As with MAIT cell activation markers, exhaustion markers LAG-3 and TIM-3 in septic patients at Day 90 had returned to levels similar to that of HD (*Figure 6E*). Of note, PD-1 expression was lower (but not statistically significant, p=0.11) in septic patients compared to HD (*Figure 6E*).

We then examined MAIT cell effector functions by analyzing the intracellular expression of MAIT-associated inflammatory cytokines (IFN-γ, IL-17, and TNF-α) and granzyme B after ex vivo TCR stimulation using *E. coli* (*Bennett et al., 2017*). Upon ICU admission, MAIT cells from septic patients showed significantly less IFN-γ production after *E. coli* stimulation, compared with HD. As with MAIT cell activation and exhaustion markers, IFN-γ production returned to levels comparable with HD at Day 90 (*Figure 7A*). No significant differences were observed in IL-17 (*Figure 7B*), TNF-α (*Figure 7C*), and granzyme B (*Figure 7D*) expression in MAIT cells among the groups.

Overall, these data demonstrate that during clinical sepsis, MAIT cell frequencies are lower, highly activated, and less capable of mounting IFN-γ responses upon stimulation. Moreover, at 90 days following sepsis, MAIT cell activation and IFN-γ responses have returned to levels comparable with matched HD.

## Discussion

Recent reports provide evidence that MAIT cells play an important role in antibacterial responses (*Le Bourhis et al., 2010*; *Ghazarian et al., 2017*). Despite this, very little is known about this unconventional T cell in sepsis. Our study reveals major alterations in activation, function, and dysfunction of MAIT cells in both clinical and experimental models of sepsis. We show that acutely during clinical sepsis, MAIT cells are highly activated, lower in frequency, and have an altered functional phenotype evidenced by reduced IFN-γ cytokine expression. Moreover, at 90 days after sepsis, these changes in MAIT cells have resolved as compared to matched healthy subjects. Using relevant animal models of sepsis, we show similarly that tissue-resident MAIT cells are dysfunctional during sepsis. Moreover, our experimental data indicate that MAIT cells are protective during sepsis, as a deficiency of MAIT cells resulted in significantly higher bacterial burden and mortality. These results indicate that

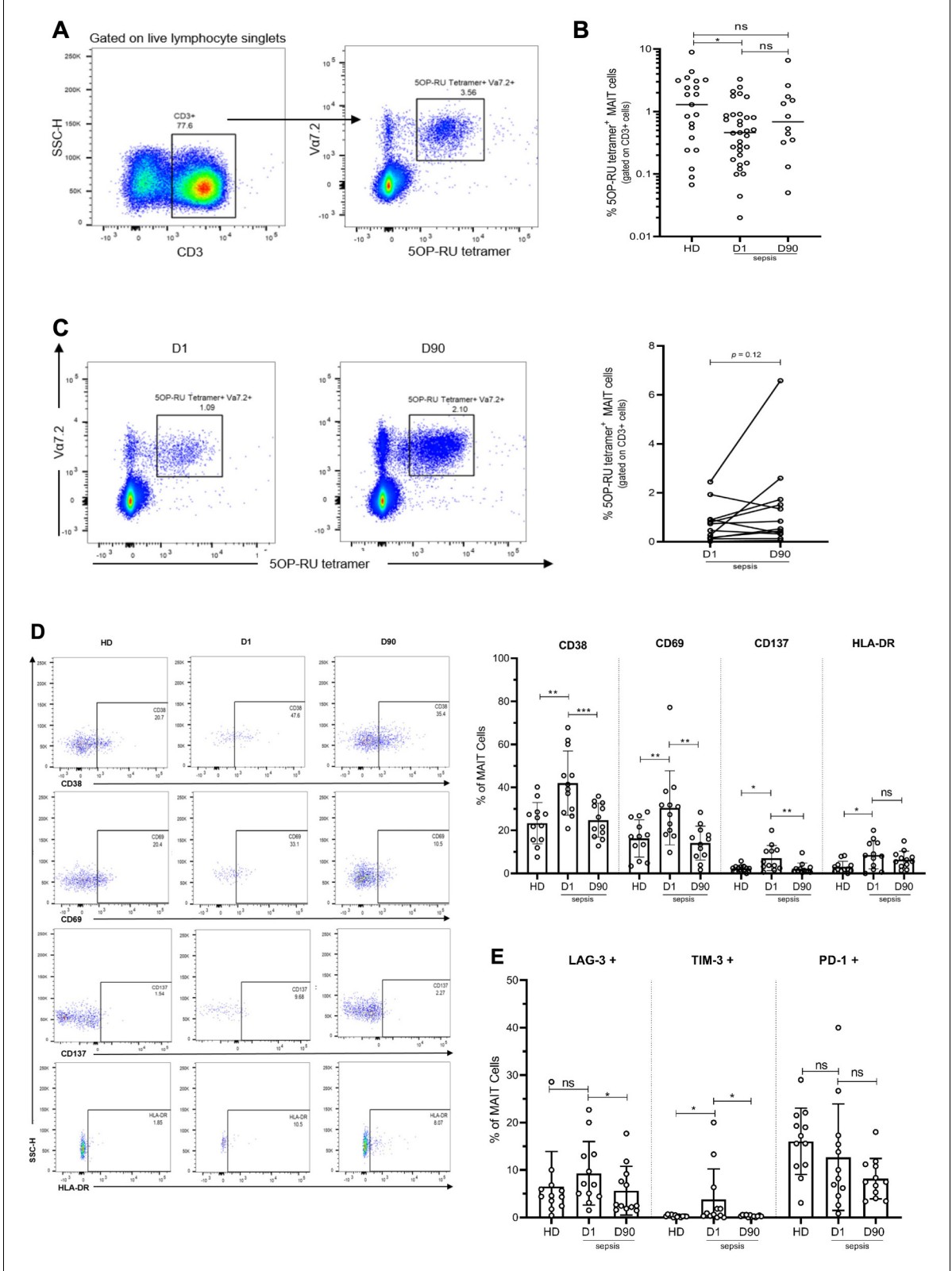

**Figure 6.** Altered frequency and phenotype of human MAIT cells in sepsis compared to healthy donors (HD). (**A**) Gating strategy for isolation of MAIT cells from PBMCs. (**B**) Percentage of MR1-tetramer+ MAIT cells in CD3+ T cells, (**C**) Representative flow plots and percentage of MAIT cells in paired Day 1 and Day 90 septic patients (n = 12 per group), (**D**) Representative flow plots and surface expression of activation markers as a percentage of MAIT cells, (**E**) Surface expression of inhibitory receptors, as a percentage of MAIT cells. Data were expressed as mean ± SD of two independent

*Figure 6 continued on next page*

*Figure 6 continued*

experiments. The Mann-Whitney test was used to compare HD with septic patients and the Wilcoxon test was used for comparisons of paired D1 and D90 samples. ***p<0.001, **p<0.01, *p<0.05. NS, p>0.05.

despite being rendered dysfunctional, MAIT cells may augment host immune responses and therefore outcomes during sepsis.

In both a polymicrobial model of sepsis (i.e. CLP) and a monomicrobial model using a clinical isolate of ExPEC, we found that MAIT-deficient ($Mr1^{-/-}$) mice have higher mortality and that this is associated with a higher bacterial burden. Our experimental data from the ExPEC model of sepsis, as well as from experiments where bedding was transferred between $Mr1^{-/-}$ and WT mice, suggest that the differences in mortality observed with CLP are unlikely to be due to differences in the microbiota. We also found that $Mr1^{-/-}$ mice had lower levels of tissue cytokines (IFN-$\gamma$, TNF-$\alpha$, IL-17, GM-

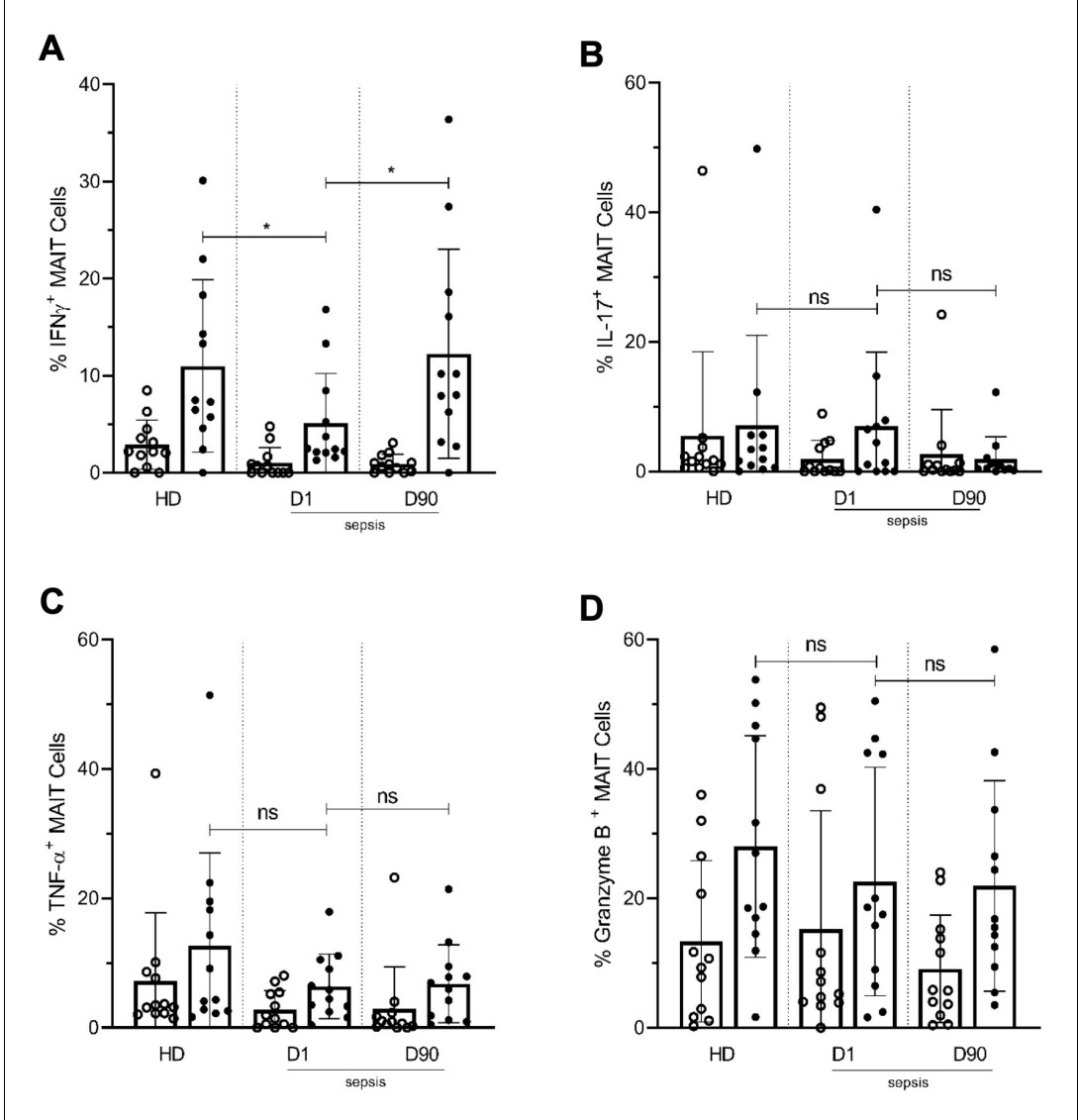

**Figure 7.** Lower frequencies of IFN$\gamma^+$ MAIT cells in day one septic patients compared to HD and Day 90 paired septic patients. PBMCs obtained from septic patients and healthy donors (n = 12 per group) were stimulated with *E. coli* at moi of 10 and intracellular expression of (**A**) IFN-$\gamma$, (**B**) IL-17, (**C**) TNF-$\alpha$, and (**D**) granzyme B by MAIT cells were analyzed using flow cytometry (open circle shows unstimulated and closed circle shows stimulated data). Data were expressed as mean ± SD of two independent experiments. The Mann-Whitney test was used for comparisons of HD with septic patients and the Wilcoxon test was used for comparisons of paired D1 and D90 samples. ***p<0.001, **p<0.01, *p<0.05. NS, p>0.05.

CSF) than that in WT mice following induction of sepsis. We hypothesize that MAIT cells play a protective role against sepsis pathology by regulating cytokine expression and maintaining a balance in immune response following sepsis to minimize tissue injury. As MAIT cells are capable of bridging the innate and adaptive immune systems, it has also been postulated that their actions are mediated in part by their interaction with macrophages or monocyte-derived dendritic cells. Interestingly, we found that in sepsis, MAIT-deficient mice produce less GM-CSF and have lower frequencies of lung interstitial macrophages and monocytic dendritic cells compared to WT mice. These observations are concordant with previous work demonstrating MAIT cell promotion of early pulmonary GM-CSF production, and differentiation of inflammatory monocytes into moDCs during pulmonary infection (*Meierovics and Cowley, 2016*). Our co-culture experiments showing that MAIT cells produce higher amounts of GM-CSF upon stimulation with the MAIT ligand 5-A-RU further confirms this hypothesis. Further work to examine whether tissue-resident MAIT cells have protective effects against sepsis pathology is warranted.

Our finding of decreased MAIT cell frequencies during severe sepsis are consistent with a previous study demonstrating that MAIT cells are decreased in patients with severe sepsis, a decrease that was not seen in other T cell populations examined, including $i$NKT and $\gamma\delta$ T cells, suggesting cell-specific, rather than global, depletion (*Grimaldi et al., 2014*). To extend this work, we found that at approximately 90 days after sepsis MAIT cell frequencies had returned to similar levels observed in matched healthy donors. Furthermore, we found that MAIT cells express elevated levels of activation markers in septic patients and that these markers also return to similar levels as in healthy donors following recovery from sepsis. The depletion of MAIT cells from circulation in septic patients is similar to what has been observed in other infections (*Leeansyah et al., 2013*; *Shaler et al., 2017*; *Leung et al., 2014*), and may be a result of apoptotic cell death, TCR internalization, or migration to peripheral tissues, although the underlying mechanisms remain to be elucidated.

A previous study had shown an association between MAIT cell frequency and protection against secondary infection in the ICU (*Grimaldi et al., 2014*). In this study, we found that in D1 sepsis patients, ex vivo stimulated MAIT cells had a depressed IFN-$\gamma$ response compared to healthy donors and patients reassessed 90 days following admission for sepsis. A similar decrease in IFN-$\gamma$ response was also seen in MAIT cells in the lung tissue of mice undergoing CLP compared to mice undergoing sham surgery. Together with an increase in expression of co-inhibitory receptors on human MAIT cells, our data suggest that during clinical sepsis, MAIT cells are dysfunctional. This may prevent MAIT cells from mounting an optimal response against microbial pathogens, a hypothesis consistent with our experimental data indicating that the loss of MAIT cells results in increased bacterial burden and mortality during sepsis. Previous work has shown that superantigens from bacterial pathogens such as *Staphylococcus spp.* and *Streptococcus spp.* can directly trigger rapid activation of MAIT cells to mount IFN-$\gamma$, TNF-$\alpha$, and IL-2 responses in an MR1-independent manner (*Shaler et al., 2017*), and that MAIT cells primed by superantigens are exhausted and anergic to cognate antigens such as *E. coli*.

Our study has several limitations. First, our co-culture experiments demonstrating the production of GM-CSF by ligand stimulation of expanded lung MAIT cells in presence of lung monocytes were performed ex vivo. Further experiments on the protective role of MAIT-derived GM-CSF during sepsis are needed. Second, our study of patients was not powered to detect differences in MAIT cells between causes of sepsis. Larger studies are needed to examine differences in MAIT cell activity and function between different causes. Finally, our experiments demonstrated decreases in lung interstitial macrophages and monocytes, along with GM-CSF production, are associated with response to CLP in $Mr1^{-/-}$ mice. But whether these deficient responses resulted in increased lung injury remains to be investigated further.

In conclusion, we demonstrate that MAIT cells undergo functional and phenotypic changes during clinical and experimental sepsis. We found that MAIT deficient mice have a higher microbial burden, lower tissue cytokine production, and higher mortality during sepsis. Despite this dysfunction, our data provide new insights into the potential protective role of MAIT cells, and their contribution toward tissue-specific cytokine responses, during sepsis. Defining MAIT cells as a critical therapeutic target during sepsis, and defining the mechanisms that regulate MAIT cell function, will potentially enable the development of novel translational strategies targeted to enhance MAIT cell functionality to improve outcomes in sepsis.

# Materials and methods

## Key resources table

| Reagent type (species) or resource | Designation | Source or reference | Identifiers | Additional information |
|---|---|---|---|---|
| Antibody | FITC anti-mouse CD3 antibody (rat monoclonal) | BioLegend | Cat# 100203 | (1:50) |
| Antibody | Brilliant Violet 510 anti-mouse NK-1.1 (mouse monoclonal) | BioLegend | Cat# 108738 | (1:50) |
| Antibody | BV711 anti-mouse CD49b (hamster monoclonal) | BD | Cat# 740704 | (1:100) |
| Antibody | PE/Cyanine7 anti-mouse TCR γ/δ (hamster monoclonal) | BioLegend | Cat# 118123 | (1:100) |
| Antibody | BV421 anti-mouse TCRβ (hamster monoclonal) | BioLegend | Cat# 109229 | (1:100) |
| Antibody | PE/Cyanine5 anti-mouse/human CD45R/B220 (rat monoclonal) | BioLegend | Cat# 103209 | (1:100) |
| Antibody | Brilliant Violet 650 anti-mouse/human CD44 (rat monoclonal) | BioLegend | Cat# 103049 | (1:100) |
| Antibody | Alexa Fluor 700 anti-mouse CD45 (rat monoclonal) | BioLegend | Cat# 103127 | (1:200) |
| Antibody | FITC anti-mouse/human CD11b (rat monoclonal) | BioLegend | Cat# 101205 | (1:200) |
| Antibody | PE/Cyanine7 anti-mouse CD11c (hamster monoclonal) | BioLegend | Cat# 117317 | (1:100) |
| Antibody | BV711 anti-mouse Siglec-F (rat monoclonal) | BD | Cat# 740764 | (1:100) |
| Antibody | Brilliant Violet 605 anti-mouse CD64 (FcγRI) (mouse monoclonal) | BioLegend | Cat# 139323 | (1:50) |
| Antibody | PE anti-mouse CD24 (rat monoclonal) | BioLegend | Cat# 101807 | (1:100) |
| Antibody | PerCP/Cyanine5.5 anti-mouse Ly-6G (rat monoclonal) | BioLegend | Cat# 127616 | (1:100) |
| Antibody | BV510 anti-mouse CD103 (hamster monoclonal) | BioLegend | Cat# 121423 | (1:50) |
| Antibody | APC anti-mouse CD86 (rat monoclonal) | BioLegend | Cat# 105011 | (1:100) |
| Antibody | PE/Dazzle 594 anti-mouse Ly-6C (rat monoclonal) | BioLegend | Cat# 128043 | (1:100) |
| Antibody | Brilliant Violet 421 anti-mouse I-A/I-E (rat monoclonal) | BioLegend | Cat# 107631 | (1:100) |
| Antibody | Alexa Fluor 700 anti-human CD137 (4-1BB) (mouse monoclonal) | BioLegend | Cat# 309816 | (1:100) |
| Antibody | BUV395 Mouse Anti-Human CD3 | BD | Cat# 563548 | (1:200) |
| Antibody | Anti-human CD8 Monoclonal Antibody (3B5), PE-Cyanine5.5 | Thermo Fisher Scientific | Cat# MHCD0818 | (1:200) |
| Antibody | BUV496 Mouse Anti-Human CD4 | BD | Cat# 612937 | (1:100) |
| Antibody | Brilliant Violet 711 anti-human TCR Vα7.2 (mouse monoclonal) | BioLegend | Cat# 351731 | (1:50) |

*Continued on next page*

*Continued*

| Reagent type (species) or resource | Designation | Source or reference | Identifiers | Additional information |
|---|---|---|---|---|
| Antibody | Brilliant Violet 785 anti-human CD223 (LAG-3) (mouse monoclonal) | BioLegend | Cat# 369321 | (1:50) |
| Antibody | Brilliant Violet 650 anti-human CD25 (mouse monoclonal) | BioLegend | Cat# 302633 | (1:100) |
| Antibody | Brilliant Violet 605 anti-human CD279 (PD-1) (mouse monoclonal) | BioLegend | Cat# 329923 | (1:100) |
| Antibody | Brilliant Violet 510 anti-human CD161 (mouse monoclonal) | BioLegend | Cat# 339921 | (1:100) |
| Antibody | PE/Cyanine5 anti-human CD69 (mouse monoclonal) | BioLegend | Cat# 310907 | (1:100) |
| Antibody | FITC anti-human HLA-DR | BioLegend | Cat# 980402 | (1:50) |
| Antibody | Brilliant Violet 421 anti-human CD366 (Tim-3) (mouse monoclonal) | BioLegend | Cat# 345007 | (1:50) |
| Antibody | PE/Cyanine7 anti-human CD38 (mouse monoclonal) | BioLegend | Cat# 303515 | (1:100) |
| Antibody | FITC anti-human IL-17A Antibody (mouse monoclonal) | BioLegend | Cat# 512303 | (1:100) |
| Antibody | PE/Cyanine7 anti-human IFN-γ (mouse monoclonal) | BioLegend | Cat# 502527 | (1:200) |
| Antibody | Anti–human TNF alpha Monoclonal Antibody (MAb11), eFluor 450 | Thermo Fisher Scientific | Cat# 48-7349-42 | (1:100) |
| Antibody | Alexa Fluor 700 anti-human/mouse Granzyme B | BioLegend | Cat# 372221 | (1:100) |

## Mice

10-14 weeks old male C57BL/6J WT mice were obtained from The Jackson laboratory and *Mr1*^-/- mice, which were back-crossed to the C57BL/6N background (10 generations) and then the C57BL/6J background (10 generations) (*Smith et al., 2019*), were obtained from Siobhan Cowley (US FDA). All mice were bred in a pathogen-free facility at the University of Utah. The animals were kept at a constant temperature (25°C) with unlimited access to a pellet diet and water in a room with a 12 hr light/dark cycle. All animals were monitored daily and infected animals were scored for the signs of clinical illness severity as previously described (*Yost et al., 2016*). Animals were ethically euthanized using $CO_2$.

## Mouse models of sepsis

C57BL/6 mice were anesthetized with ketamine/xylazine (100 mg/kg and 10 mg/kg, i.p., respectively), and sepsis was induced by cecum ligation and puncture (CLP) as previously described (*Hubbard et al., 2005*). After surgery, the animals received subcutaneous sterile isotonic saline (1 mL) for fluid resuscitation. Sham-operated mice were subjected to identical procedures except that CLP was not done. The animals were followed for 5 days after the surgical procedure for the determination of survival. Bacterial loads were determined by serial dilution and plating of lung homogenates and blood on LB agar plates.

To examine monomicrobial sepsis, mice were injected intraperitoneally with 200 μL of PBS containing ~1×10^6 CFU of extraintestinal pathogenic *Escherichia coli* (ExPEC) clinical strain F11 (*Smith et al., 2007*). Bacteria (F11) was grown from frozen stocks in static modified M9 media at 37°C for 24 hr before use in the sepsis model. Mice were monitored for 5 days for determination of survival.

## Bedding transfer procedure

The bedding transfer procedure was followed as previously described (*Miyoshi et al., 2018*). Briefly, bedding transfers were performed among $Mr1^{-/-}$ cages and WT cages. The bedding was mixed at 3–4 days and at 8–10 days following these cage changes (as fresh bedding is provided every 14 days, this represents bedding transfers performed twice within each 2-week cycle). At each of these time points, roughly one-quarter of soiled bedding was collected from each cage and the bedding from all cages was mixed in an autoclaved sterile container, followed by redistribution across all cages. Soiled bedding was collected and mixed within a freshly cleaned biological safety cabinet within the room where animals were housed. After 4–5 weeks of bedding transfers, polymicrobial sepsis was induced using CLP.

## Lung mononuclear cell isolation

For lung digestion and preparation of single cell suspensions, the lungs were perfused using sterile phosphate-buffered saline. After removal, the lung dissociation protocol (Miltenyi Biotec, Bergisch Gladbach, Germany) was performed using the mouse Lung Dissociation Kit (Miltenyi Biotec) and the gentleMACS Dissociator (Miltenyi Biotec) as per manufacturer's instructions. After dissociation, cells were passed through a 70 μm cell strainer and washed with RPMI with 10% FBS. Red blood cells were lysed with red blood cell lysis buffer. Lung mononuclear cells were then washed twice in RPMI with 10% FBS before use in subsequent experiments.

## Tetramer and surface-staining of lung single cell suspensions

From each group of animals, 1–2 million cells aliquots were prepared and to exclude dead cells from analysis, cells were first stained with the fixable viability dye eFluor 780 (eBioscience) for 15 min at room temperature (RT). Cells were incubated with anti-mouse CD16/CD32 Fc Block antibody (BD Biosciences, San Jose, CA), for 20 min at 4°C. Cells were then stained for 30 min at RT with appropriately diluted PE-conjugated 5-OP-RU-loaded species-specific MR1-tetramers or α-GalCer (PBS-44)–loaded CD1d tetramer conjugated to APC, anti-CD3-FITC (BioLegend), anti-CD161-BV510 (BioLegend), anti-CD49b-BV711 (BD), anti-TCRγδ-PE-Cy7 (BioLegend), anti-TCRβ-BV421 (BioLegend), anti-CD45R-PE-Cy5, and anti-CD44-BV650 (BioLegend). To evaluate different antigen-presenting cells, after Fc Block incubation, cells were also surface stained with anti-CD45 AF700 (BioLegend), anti-CD11b-FITC (BioLegend), anti-CD11c-PE Cy-7 (BD), anti-Siglec-F- BV711 (BD), anti-CD64-BV605 (BioLegend), anti-CD24-PE (BioLegend), anti-Ly-6G-PerCP-Cy5.5 (BioLegend), anti-CD103-BV510 (BioLegend), anti-CD86-APC (BioLegend), anti-Ly6C (PE dazzle 594), and anti-MHC II BV421 (BioLegend) for 30 min at 4°C. Total $10^6$ gated events per sample were collected using Fortessa flow cytometer (Becton Dickinson, San Diego, CA), and results were analyzed using FlowJo 10.4.2 software.

## Quantitation of MAIT cytokine transcripts by real-time PCR

MAIT cells ($CD3^+$ $CD44^+$ $TCR\beta^+$ MR1-5-OP-RU tetramer$^+$) and non-MAIT (MR1-5-OP-RU tetramer$^-$ $TCR\beta^+$) $CD3^+$ T cell populations were sorted into 350 μL RLT plus buffer from the Qiagen Allprep DNA/RNA micro kit. Nucleic acid extraction was performed according to manufacturer's instructions (Qiagen Allprep RNA/DNA). cDNA was synthesized using SuperScript VILO cDNA Synthesis Kit and qRT-PCR analysis on *Ifng*, *Il17a*, *Gzmb,* and *Prf1* genes was performed with specific TaqMan primers and TaqMan Fast Advanced Master Mix (Applied Biosystems). The mRNA levels of specific genes were determined by the relative standard curve method, normalized against housekeeping ribosomal protein L32 levels. ΔCT was calculated as $CT_{gene} - CT_{housekeeping}$ and ΔΔCT was calculated as $\Delta CT_{CLP} - \Delta CT_{sham}$.

## Mouse cytokines

Following lung dissociation, cells were passed through a 70 μm nylon mesh. Cells and debris were removed from the suspension via centrifugation at $300 \times g$ for 10 min and supernatant were collected and stored at −80°C for later use. Lung cytokine levels were assessed from the supernatant samples via LEGENDplex (mouse inflammation panel 13-plex; BioLegend) kit per manufacturer's instructions. Serum samples were 2-fold diluted and cytokine levels were assessed using the same kit as per manufacturer's instructions. Cytokine levels were acquired using a FACSCanto II flow

cytometer (BD Biosciences), and analyses were performed using LEGENDplex data analysis software (BioLegend).

## Ex vivo co-culture of lung Ly6C$^+$ CD11b$^+$ monocytes with MAITs

Lung MAITs were expanded ex vivo using a previously published method for the expansion of human MAIT cells (Liu et al., 2020). Briefly, lung homogenates from healthy WT mice were cultured for 14 days in the presence of mouse recombinant IL-2 and sulfate latex beads of 5-OP-RU/MR1 artificial antigen-presenting cells. To isolate monocytes, lung cells from healthy WT mice were surface stained with anti-CD45 AF700 (BioLegend), anti-CD11b-FITC (BioLegend), anti-Siglec-F- BV711 (BD), and anti- Ly6C (PE dazzle 594) for 30 min at 4°C. We obtained CD45$^+$, Siglec-F$^-$, Ly6C$^+$, CD11b$^+$ monocytes by flow sorting on a FACS Aria II (BD, Franklin Lakes, NJ). We incubated 20,000 MAITs with 10,000 monocytes for 18 hr, in a total volume of 200 µL in 96-well U-bottom plates. MAIT-monocyte co-culture was stimulated with 200 nM 5-A-RU and 50 µM MeG (Li et al., 2018) in RPMI 1640 with 10% FBS with 1% penicillin/streptomycin and 1% HEPES. MAIT cells that were stimulated/unstimulated and monocytes that were stimulated/unstimulated were kept as controls. After 18 hr, the cell supernatant was collected and stored at −20°C. We measured IFN-γ and GM-CSF levels in supernatants using ELISA as per manufacturer's instruction in BioLegend ELISA MAX standard sets for mouse IFN-γ and mouse GM-CSF.

## Human subjects

Patients admitted to an academic medical intensive care unit (ICU) with a primary diagnosis of severe sepsis or septic shock were prospectively enrolled within 48 (±24) hr of ICU admission. Sepsis was defined using the consensus criteria (Bone et al., 1992; American College of Chest Physicians/ Society of Critical Care Medicine Consensus Conference, 1992) at the time this study was actively recruiting, defined as suspected or confirmed systemic infection and organ dysfunction as defined by a sequential organ failure assessment (SOFA) score ≥2 above the baseline. Septic shock was defined as sepsis and an elevated lactate >2 mmol/L or the need for vasopressors. Blood samples were collected from septic patients upon study enrollment (Day 1, after ICU admission for sepsis) and again in the same subjects ~90 days after enrollment (Day 90). The venous blood was collected and centrifuged over a Ficoll-Hypaque density gradient using the standard protocol to isolate peripheral blood mononuclear cells (PBMCs), which were either cryopreserved or used directly. The samples from healthy donors were collected and processed following the same procedure.

## Analysis of MAIT cytokine expression by flow cytometry

For phenotypic analysis of MAIT cells, PBMCs were thawed from −80°C and were stained for surface markers: anti-CD137-Alexa Fluor 700 (BioLegend), anti-CD3-BUV395 (BD Biosciences), anti-CD8-PE-Cy5.5 (Molecular Probes), anti-CD4-BUV496 (BD Biosciences), anti-Vα7.2-BV711 (BioLegend), anti-LAG3-BV785 (BioLegend), anti-CD25-BV650 (BioLegend), anti-PD-1-BV605 (BioLegend), anti-CD161-BV510 (BioLegend), anti-CD69-PE-Cy5 (BioLegend), anti-HLA-DR-FITC (BioLegend), anti-TIM3-BV421 (BioLegend), anti-CD38-PE-Cy7 (BioLegend), and anti-human MR1 5-OP-RU Tetramer (NIH Tetramer Core Facility). PBMCs that were cultured and stimulated with 1100–2 E. coli (E. coli Genetic Stock Center [CGSC], Yale University) at a multiplicity of infection (MOI) of 10 for 18 hr were stained for surface markers: anti-human MR1 5-OP-RU Tetramer (NIH Tetramer Core Facility), anti-CD3-BUV395 (BD Biosciences), anti-CD8-PE-Cy5.5 (Molecular Probes), anti-CD4-BUV496 (BD Biosciences), anti-Vα7.2-BV711 (BioLegend), anti-LAG3-BV785 (BioLegend), anti-CD25-BV650 (BioLegend), anti-PD-1-BV605 (BioLegend), anti-CD161-BV510 (BioLegend), anti-CD69-PE-Cy5 (BioLegend) and intracellularly stained for: anti-IL-17A-FITC (BioLegend), anti-IFN-γ-PE-Cy7 (BioLegend), anti-TNFα-ef450 (Molecular Probes), anti-Granzyme B-Alexa Fluor 700 (BioLegend). All samples were acquired using a 5-laser BD LSRFortessa II flow cytometer (BD Biosciences) and analyzed using FlowJo software v10 (Tree Star Inc, Ashland, OR). Unstimulated controls were used to define a positive population.

## Statistics and power calculations

Comparisons between independent groups were performed with unpaired t-test. The Mann-Whitney U-test was used for comparison of continuous variables between two groups and results of lung

cytokine analysis were presented as mean ± standard deviation. For comparisons of more than two groups, one-way ANOVA with Tukey's multiple comparison test was used.

For comparisons of healthy subjects with day 1 or day 90 septic patients, the Mann-Whitney U-test was used and for comparisons of paired day 1 and day 90 samples, the Wilcoxon test was used. GraphPad Prism 8.3.0 software was used for all statistical analyses and $p<0.05$ was considered statistically significant.

We used the Cox proportional-hazards (coxph) model with non-bedding-transferred mice as the reference for determining power calculation for the number of mice needed in the bedding transfer experiment. The formula used was: coxph(formula = Surv(t, event)~group, data = exp) with coef = 2.1525, exp(coef)=8.6065, se(coef) = 0.7724, z = 2.787, and p group = 0.00532. The likelihood ratio test = 11.35 on one df, p = 0.0007563, n = 30, number of events = 13. In order to observe the hazard ratio from this study (8.6065) with 80% power and type I error 0.05, we need to observe seven deaths. In order to determine the sample size, we divided seven by the expected rate of death overall. For an expected death rate of 50%, we needed a sample size of 14; for an expected death rate of 25%, we needed a sample size of 28.

## Study approval

Each patient or a legally authorized representative provided the written informed consent. The Institutional Review Board approved this study. All animals were maintained and experiments were performed in accordance with The University of Utah and Institutional Animal Care and Use Committee (IACUC) approved guidelines (protocol # 18–10012).

## Acknowledgements

This research was supported by the National Institutes of Health (AI130378 to DTL, HL092161, AG040631, and AG048022 to MTR, TL1TR002540 to DL, T32 HG008962 to CPA), and a University of Utah 3i Initiative Seed Grant (to DTL, MTR, JSH). We would like to thank all the study subjects who participated in the study. We thank Kelin Li and Jeffrey Aube for providing 5-amino-6-D-ribitylaminouracil (5-A-RU) for our study. We would like to thank Michael C Graves and Alexandra Heitkamp for their help in the laboratory and Ben Brintz, Study Design and Biostatistics Center, University of Utah, for help with statistics and power calculations. We would also like to thank the staff of the University of Utah Flow Cytometry Core and the Office of Comparative Medicine.

## Additional information

### Funding

| Funder | Grant reference number | Author |
| --- | --- | --- |
| National Institute of Allergy and Infectious Diseases | AI130378 | Daniel T Leung |
| National Heart, Lung, and Blood Institute | HL092161 | Matthew T Rondina |
| National Institute on Aging | AG040631 | Matthew T Rondina |
| National Institute on Aging | AG048022 | Matthew T Rondina |
| National Center for Advancing Translational Sciences | TL1TR002540 | Daniel Labuz |
| National Institute of General Medical Sciences | HG008962 | Cole P Anderson |
| University of Utah | 3i Initiative Seed Grant | J Scott Hale Matthew T Rondina Daniel T Leung |

The funders had no role in study design, data collection and interpretation, or the decision to submit the work for publication.

## Author contributions

Shubhanshi Trivedi, Formal analysis, Validation, Investigation, Visualization, Writing - original draft, Writing - review and editing; Daniel Labuz, Investigation, Visualization, Writing - review and editing; Cole P Anderson, Claudia V Araujo, Elizabeth A Middleton, Owen Jensen, Investigation, Writing - review and editing; Antoinette Blair, Data curation, Investigation, Writing - review and editing; Alexander Tran, Investigation; Matthew A Mulvey, Resources, Supervision, Investigation, Writing - review and editing; Robert A Campbell, Conceptualization, Supervision, Investigation, Project administration, Writing - review and editing; J Scott Hale, Conceptualization, Funding acquisition, Writing - review and editing; Matthew T Rondina, Conceptualization, Supervision, Funding acquisition, Project administration, Writing - review and editing; Daniel T Leung, Conceptualization, Resources, Supervision, Funding acquisition, Writing - original draft, Project administration, Writing - review and editing

## Author ORCIDs

Daniel T Leung https://orcid.org/0000-0001-8401-0801

## Ethics

Human subjects: Each patient or a legally authorized representative provided written, informed consent. The University of Utah Institutional Review Board approved this study (protocol #102638).
Animal experimentation: This study was performed in strict accordance with the recommendations in the Guide for the Care and Use of Laboratory Animals of the National Institutes of Health. All animals were maintained and experiments were performed in accordance with The University of Utah and Institutional Animal Care and Use Committee (IACUC) approved guidelines (protocol # 18-10012).

## Decision letter and Author response

Decision letter https://doi.org/10.7554/eLife.55615.sa1
Author response https://doi.org/10.7554/eLife.55615.sa2

## Additional files

### Supplementary files

• Transparent reporting form

### Data availability

All data generated or analysed during this study are included in the manuscript and supporting files.

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
