## [Decision Letter]

**Acceptance summary:**

Trivedi et al. study the role of MAIT cells during sepsis using the combined approach of assessing clinical data alongside modelling sepsis in a mouse model lacking MAIT cells. The main conclusion drawn is that MAIT cells are protective, with evidence provided by the observation that MR1 KO mice are much more susceptible to sepsis induced death compared to WT mice.

**Decision letter after peer review:**

Thank you for submitting your article "Mucosal-associated invariant T (MAIT) cells mediate protective host responses in sepsis" for consideration by *eLife*. Your article has been reviewed by three peer reviewers, including Nicola L Harris as the Reviewing Editor and Reviewer #1, and the evaluation has been overseen by Satyajit Rath as the Senior Editor.

The reviewers have discussed the reviews with one another and the Reviewing Editor has drafted this decision to help you prepare a revised submission.

Summary:

Trivedi et al. study the role of MAIT cells during sepsis using clinical samples as well as a mouse model system. The main conclusion drawn is that MAIT cells are protective, which is largely based on the observation that MR1 ko mice are much more susceptible to sepsis induced death compared to WT mice. The authors address a very relevant question and the combined approach of using clinical data and a mouse model system has the potential to be very powerful. However all reviewers agree that a number of possible caveats need to be addressed by further experimental work to fully justify the conclusions.

Essential revisions:

1) The authors should confirm the differences observed in the MR1 deficient and WT mice are not due to differences in genotype (provide confirmation of backcrossing or SNP analysis) or microbiota (repeat experiments using littermates or co-housing)

2) The authors should further investigate whether MAIT cells are indeed the main source of protective cytokines in the tissues of septic mice by performing intra-cellular cytokine staining within various cell populations or ELISA analyses of purified MAIT cells.

3) The authors argue that MAIT cells are protective and suggest that the differences in cytokine levels measured in the lungs of WT and MR1 mice are due to MAIT cell function (further proof of this was asked above in point 2). When and how are MAIT cells exerting their protective function. Do the authors suggest that MAIT cells are functional for a brief period of time (less than 18hrs), which is sufficient to elicit a protective effect? In this context, strengthening the link between MAIT cells in the lung and the monocyte accumulation in the lung would strengthen the possible mechanisms by which MAIT cells are protective and could be achieved by investigating levels of the relevant chemokines and comparing MAIT cell responses and monocyte responses over a fuller time course.

4) It is not clear why the authors chose the lungs to evaluate CLP-induced inflammation as the role of lung injury in mortality in this model is controversial. Thus justification for why only lung MAIT cells were studies (as opposed to lung, liver and spleen) should be given or the dataset expanded to include other organs. The authors should also provide evidence of lung injury that may explain the increased susceptibility of *Mr1^-/-^* mice (neutrophil levels, histology, vascular leakage).

[Editors' note: further revisions were suggested prior to acceptance, as described below.]

Thank you for submitting your article "Mucosal-associated invariant T (MAIT) cells mediate protective host responses in sepsis" for consideration by *eLife*. Your article has been reviewed by three peer reviewers, including Nicola L Harris as the Reviewing Editor and Reviewer #1, and the evaluation has been overseen by Satyajit Rath as the Senior Editor.

The reviewers have discussed the reviews with one another and the Reviewing Editor has drafted this decision to help you prepare a revised submission.

Summary:

Trivedi et al. study the role of MAIT cells during sepsis using clinical samples as well as a mouse model system. The main conclusion drawn is that MAIT cells are protective, which is largely based on the observation that MR1 ko mice are much more susceptible to sepsis induced death compared to WT mice. The authors address a very relevant question and the combined approach of using clinical data and a mouse model system has the potential to be very powerful. Although the reviewers recognize the efforts made to address the reviewers previous comments, two issues remain unresolved.

Essential revisions:

The first, and most important is that microbiota differences are likely to exist between any two independently bred strains and could easily account for the observed effects. The reviewers agree that this possibility precludes publication at this point and that experiments using littermate controls or co-housed animals would need to be performed to validate the manuscript's conclusions. The second is that further confirmation that the phenotype is due to the lack of MR1 and not other genetic differences between different C57Bl/6 strains is required. This needs to be addressed more thoroughly, preferably by performing genetic testing comparing the MR1 ^-/-^. wildtype controls and C67BL6J mice from Jax.

[Editors' note: further revisions were suggested prior to acceptance, as described below.]

Thank you for submitting your article "Mucosal-associated invariant T (MAIT) cells mediate protective host responses in sepsis" for consideration by *eLife*. Your article has been discussed by the all original peer reviewers, and the evaluation has been overseen by a Reviewing Editor and Satyajit Rath as the Senior Editor.

The reviewers have discussed the reviews with one another and the Reviewing Editor has drafted this decision to help you prepare a revised submission.

Summary

Trivedi et al. study the role of MAIT cells during sepsis using clinical samples as well as a mouse model system. The main conclusion drawn is that MAIT cells are protective, which is largely based on the observation that MR1 ko mice are much more susceptible to sepsis induced death compared to WT mice. The authors address a very relevant question and the combined approach of using clinical data and a mouse model system has the potential to be very powerful. Although the reviewers recognize the efforts made to address the reviewers previous comments, the issues of microbial influences remains unresolved and need to be clarified further with extensive work before any recommendation regarding publication can be made.

Essential revision:

The data provided by the authors in which co-housing was used and the authors report that they found a trend of increased sepsis related mortality in Mr1 KO compared to WT mice. The reviewer’s agree that they cannot make this comment based on the lack of significance (p 0.117), but that the study may lack sufficient power. If they chose the authors could perform power calculations to determine the number of mice that would need to be compared to reach a robust conclusion. Any data generated would then need to be included in the main text of the manuscript and publication would be dependent on how the outcome of such work impacts on the other conclusions drawn in the manuscript.

---

## [Author Response]

Essential revisions:1) The authors should confirm the differences observed in the MR1 defieicnt and WT mice are not due to differences in genotype (provide confirmation of backcrossing or SNP analysis) or microbiota (repeat experiments using littermates or co-housing)

Regarding confirmation of backcrossing, we used first generation mice from MR1 knockout breeding pairs provided by collaborators who had backcrossed to the B6J background (10 generations). Regarding the potential role of the microbiota, we appreciate this excellent question and recognize that microbiota can influence host susceptibility to infection. Unfortunately, we will not be able to perform additional experiments on the microbiota, given current COVID-19 related restrictions on research. We have thus added the following statement of limitations in the Discussion section, “Our study has a number of limitations. The microbiota associated with *MR1^-/-^* mice may differ from wild-type mice (37), and we have not examined whether such differences affects susceptibility to sepsis. Our experiments using ExPEC suggest the differences in mortality observed with CLP are not due to differences in the microbiota. However, future studies examining the contribution of MAIT cells to differences in microbiota, and their effects on susceptibility to sepsis, are needed.”

2) The authors should further investigate whether MAIT cells are indeed the main source of protective cytokines in the tissues of septic mice by performing intra-cellular cytokine staining within various cell populations or ELISA analyses of purified MAIT cells.

We appreciate this suggestion and agree. To investigate whether MAIT cells are indeed the main source of protective cytokines in the tissues of septic mice, we performed new experiments with intra-cellular cytokine staining within various cell populations. We found that percentage frequencies of IFNγ in lung MAIT cells were significantly lower in septic mice compared to sham treated mice. Conversely, in non-MAIT TCRβ+CD3+ T cells, frequencies of IFNγ+ cells were higher in septic mice compared to sham treated mice. No significant differences were observed in frequencies of TNFα, IL-17a, GM-CSF and IL-10 in lung MAIT cells between the groups. No differences were observed in iNKT and TCRγδ IFNγ production.

We have now added these results as a new Figure 2 and have added to the Results section to state, “To further confirm that MAIT cell effector function is impaired during experimental sepsis, we evaluated IFNγ, TNFα, IL-17a, GM-CSF and IL-10 protein expression in MAIT cells of lung tissue using flow cytometry. We found that similar to *IFNγ* mRNA expression, percentage frequencies of MAIT cells expressing IFNγ were significantly lower (P = 0.04) in septic mice compared to sham mice (Figure 2A). Mean fluorescent intensity (MFI) of IFNγ staining in TCRβ^+^ MR1-5OP-RU-tetramer^+^ MAIT cells was also significantly reduced (P = 0.02) in septic mice compared to sham mice (Figure 2B). Conversely, in non-MAIT TCRβ+CD3+ T cell populations, IFNγ protein expression was significantly higher (P = 0.0005) in septic mice compared to sham mice (Figure 2C). No significant differences were observed in IFNγ expression in iNKT and TCRγδ cells (Figure 2D and E). No significant differences were observed in frequencies of TNFα, IL-17a, GM-CSF and IL-10 between the groups.”

3) The authors argue that MAIT cells are protective and suggest that the differences in cytokine levels measured in the lungs of WT and MR1 mice are due to MAIT cell function (further proof of this was asked above in point 2). When and how are MAIT cells exerting their protective function. Do the authors suggest that MAIT cells are functional for a brief period of time (less than 18hrs), which is sufficient to elicit a protective effect? In this context, strengthening the link between MAIT cells in the lung and the monocyte accumulation in the lung would strengthen the possible mechanisms by which MAIT cells are protective and could be achieved by investigating levels of the relevant chemokines and comparing MAIT cell responses and monocyte responses over a fuller time course.

We agree and tested the hypothesis that MAIT cells exert their protective function by IFNγ and GM-CSF cytokine expression very early following sepsis. Total MAIT cells in the lung of each mouse number only ~1000, making co-culture experiments difficult. Thus, we expanded lung MAITs ex vivo using a published method for expansion of human MAITs using recombinant IL-2 and sulfate latex beads of 5-OP-RU/MR1 artificial antigen-presenting cells for 14 days (Liu et al., 2020). This resulted in a 10 fold increase in MAIT cell numbers. We then co-cultured these expanded MAITs with Ly6C+ CD11b+ monocytes isolated from lungs of WT mice for 18 hrs. We found significantly higher amounts of IFNγ and GM-CSF in co-cultures stimulated with 5-A-RU/MeG compared to unstimulated and monocyte only controls. This suggests that within 18 hours, MAITs and monocytes interact in the lung and produce relevant cytokines.

We have presented these results as supplementary figures to Figure 5 and added to the Results section, “To investigate the possible mechanism by which MAIT cells are protective, we performed ex vivo expansion of lung MAIT cells from WT mice using 5-OP-RU/MR1 artificial antigen presenting cells (25) and co-cultured them with Ly6C+ CD11b+ monocytes, also isolated from WT mice, for 18 hours. We found significantly higher amounts of IFNγ (Figure 5—figure supplement 1A) and GM-CSF (Figure 5—figure supplement 1B) in co-cultures stimulated with 5-amino-6-D-ribitylaminouracil (5-A-RU) and methylglyoxal (MeG) (25) compared to unstimulated and monocyte only controls indicating that MAIT cells in lung can interact with monocytes to produce effector cytokines.”

We have also added to the Discussion section, “Our co-culture experiments showing that MAIT cells produce higher amounts of GM-CSF upon stimulation with the MAIT ligand 5-A-RU further confirms this hypothesis.” We have also added to the Discussion section, “Secondly, our coculture experiments demonstrating the production of GM-CSF by ligand stimulation of expanded lung MAIT cells in presence of lung monocytes were ex vivo. Further experiments on the protective role of MAIT-derived GM-CSF during sepsis are needed.*”*

We have also added to the Materials and methods section, “Ex vivo co-culture of lung Ly6C+ CD11b+ monocytes with MAITs: Lung MAITs were expanded ex vivo using a previously published method for expansion of human MAIT cells (25). Briefly, lung homogenates from healthy WT mice were cultured for 14 days in presence of mouse recombinant IL-2 and sulfate latex beads of 5-OP-RU/MR1 artificial antigen-presenting cells. To isolate monocytes, lung cells from healthy WT mice were surface stained with anti-CD45 AF700 (Biolegend), anti-CD11b-FITC (Biolegend), anti-Siglec-F- BV711 (BD), and anti- Ly6C (PE dazzle 594) for 30 min at 4 °C. We obtained CD45+ Siglec-F- Ly6C+ CD11b+ monocytes by flow sorting on a FACS Aria II (BD, Franklin Lakes, NJ, USA). We incubated 20,000 MAITs with 10,000 monocytes for 18 hours, in a total volume of 200µl in a 96-well U-bottom plates. MAIT-monocyte coculture was stimulated with 200nM 5-A-RU and 50µM MeG (26) in RPMI 1640 with 10% FBS with 1% penicillin/streptomycin and 1% HEPES. MAIT cells that were stimulated/unstimulated and monocytes that were stimulated/unstimulated were kept as controls. After 18 hours, cell supernatant was collected and stored at -20°C. We measured IFN-γ and GM-CSF levels in supernatants using ELISA as per manufacturer’s instruction in Biolegend ELISA MAX standard sets for mouse IFN-γ and for mouse GMCSF.”

4) It is not clear why the authors chose the lungs to evaluate CLP-induced inflammation as the role of lung injury in mortality in this model is controversial. Thus justification for why only lung MAIT cells were studies (as opposed to lung, liver and spleen) should be given or the dataset expanded to include other organs. The authors should also provide evidence of lung injury that may explain the increased susceptibility of Mr1^-/-^ mice (neutrophil levels, histology, vascular leakage).

We apologize for any confusion, and appreciate the chance to clarify. MAIT cells are most abundant in lungs as compared to other organs in mice (Rahimpour et al., 2015). Nevertheless, to examine other tissues, we evaluated MAIT-associated cytokines in lung, liver, gut and serum after CLP. We have now inserted (added as Fiugre 5—figure supplement 1 and 2, also in Results section) additional data showing that tissue cytokines in serum, liver and gut were less affected than lung in *MR1^-/-^* mice after CLP. We did not observe significant differences in frequencies of lung neutrophils between *MR1^-/-^*and WT mice, and have added to Figure 5 a new subpanel (panel E) showing these data. We have also updated the Results section as: “Loss of MAIT cells did not influence AMs, pDCs, or neutrophils, however, following CLP (Figure 5C -E)”. Unfortunately, due to COVID-19 related restrictions, we were unable to generate additional data regarding the phenotype of lung injury. We have added additional statement of limitations to the Discussion, “Lastly, our experiments demonstrated decreases in lung interstitial macrophages and monocytes, along with GM-CSF production, are associated with response to CLP in *MR1^-/-^* mice. But whether these deficient responses resulted in increased lung injury remains to be investigated further.”

[Editors' note: further revisions were suggested prior to acceptance, as described below.]

Essential revisions:The first, and most important is that microbiota differences are likely to exist between any two independently bred strains and could easily account for the observed effects. The reviewers agree that this possibility precludes publication at this point and that experiments using littermate controls or co-housed animals would need to be performed to validate the manuscript's conclusions.

Thank you for this suggestion. To address this, we used a bedding transfer protocol as published in Miyoshi et al., 2018. Briefly, soiled cage bedding was routinely mixed weekly and distributed among all WT and *Mr1^-/-^* mice cages for total of four weeks followed by CLP. Similar to our CLP model and ExPEC model results, we found a trend of increased sepsis-related mortality in *Mr1^-/-^* mice compared to WT mice. The majority (7/10) of *Mr1^-/-^* mice died in 48-72 hours following induction of sepsis, while the majority (6/10) of WT mice survived up to 100 hours after sepsis (Log-rank test, *p* 0.117).

**Author response image 1. sa2fig1:** 

The second is that further confirmation that the phenotype is due to the lack of MR1 and not other genetic differences between different C57Bl/6 strains is required. This needs to be addressed more thoroughly, preferably by performing genetic testing comparing the MR1 +. wildtype controls and C67BL6J mice from Jax.

The MR1 null mice we received from our collaborator at the FDA were 10x backcrossed prior to us receiving it. We have cited the publications of the collaborator who provided us with the mice, (Smith et al., 2019, “The MR1^-/-^ mice were originally generated on a mixed B6/129OlaHsd background. The mice were back-crossed to the C57BL/6N background (10 generations) and then the B6J background (10 generations).”). We have now added this in the Materials and methods: “Mr1^-/-^ mice, which were back-crossed to the C57BL/6N background (10 generations) and then the C57BL/6J background (10 generations) (37), were obtained from Siobhan Cowley (US FDA).” In addition, in this paper, we report that there were no differences in many immune cells examined between control wild-type and MR1 null mice under sham surgery conditions.

[Editors' note: further revisions were suggested prior to acceptance, as described below.]

Essential revision:The data provided by the authors in which co-housing was used and the authors report that they found a trend of increased sepsis related mortality in Mr1 KO compared to WT mice. The reviewer’s agree that they cannot make this comment based on the lack of significance (p 0.117), but that the study may lack sufficient power. If they chose the authors could perform power calculations to determine the number of mice that would need to be compared to reach a robust conclusion. Any data generated would then need to be included in the main text of the manuscript and publication would be dependent on how the outcome of such work impacts on the other conclusions drawn in the manuscript.

We thank reviewers for their suggestions. We have now done power calculations and repeated this experiment, for total of 3 independent experiments, with total n = 31 *Mr1*^-/-^mice and n = 30 WT mice. We found significant increase in *Mr1*^-/-^ mortality compared to WT mice (p = 0.03). Together with existing experimental data from an alternative model of sepsis (ExPEC injection), which showed similar findings, we conclude that differences in mortality observed with CLP are not due to differences in the microbiota.

Following changes are made in manuscript.

Results section: We have added Figure 3C, and revised the text to state, “The microbiota associated with *Mr1^-/-^* mice may differ from wild-type mice (22); therefore we examined whether such differences affect susceptibility to sepsis. In experiments in which bedding was transferred between *Mr1^-/-^* and WT cages (23) prior to CLP, we found that *Mr1^-/-^* mice had significantly increased mortality compared to WT mice (Figure 3C).”

Materials and methods section:

Bedding transfer procedure is added: “Bedding transfer procedure was followed as previously described (23). Briefly, bedding transfers were performed among *Mr1^-/-^* and WT cages. Bedding was mixed at 3–4 days and at 8–10 days following cage changes (as fresh bedding is provided every 14 days, this represents bedding transfers performed twice within each two-week cycle). At each of these time points, roughly one-quarter of soiled bedding was collected from each cage and the bedding from all cages was mixed in an autoclaved sterile container, followed by redistribution across all cages. Soiled bedding was collected and mixed within a freshly cleaned biological safety cabinet within the room where animals were housed. After 4-5 weeks of bedding transfers, polymicrobial sepsis was induced using CLP.”

Power calculations are added:

“We used Cox proportional-hazards (coxph) model with non-bedding-transferred mice as reference for determining power calculation for the number of mice needed in the bedding transfer experiment. Formula used was: coxph(formula = Surv(t, event) ~ group, data = exp) with coef = 2.1525, exp(coef) = 8.6065, se(coef) = 0.7724, z = 2.787 and p group = 0.00532. Likelihood ratio test = 11.35 on 1 df, p=0.0007563, n= 30, number of events= 13. In order to observe the hazard ratio from this study (8.6065) with 80% power and type I error 0.05, we need to observe 7 deaths. In order to determine sample size, we divided 7 by the expected rate of death overall. For an expected death rate of 50%, we needed a sample size of 14; for an expected death rate of 25%, we needed a sample size of 28. We used total 31 *Mr1^-/-^* and 30 WT mice from three independent experiments to plot survival curves.”

Discussion section: We have added the following text to the second paragraph of the Discussion: “Our experimental data from the ExPEC model of sepsis, as well as from experiments where bedding was transferred between *Mr1^-/-^* and WT mice, suggest that the differences in mortality observed with CLP are unlikely to be due to differences in the microbiota.”

Figure legend for Figure 3C is added as: “(C) Bedding transfer procedure (detailed in the Materials and methods section) was used to exchange gut microbiome between age matched *Mr1^-/-^* and WT mice (total n = 31 for *Mr1^-/-^* mice and total n = 30 for WT mice) before inducing sepsis by CLP. Survival was recorded over a period of 4-5 days (WT versus *Mr1^-/-^* mice, * p = 0.03). Data represents three independent experiments. Statistical analysis was performed using Log-rank (Mantel-Cox) test.”